# Photodegradation of Bisphenol a in Water via Round-the-Clock Visible Light Driven Dual Layer Hollow Fiber Membrane

Khalis Sukaini [1], Siti Hawa Mohamed Noor [1], Sumarni Mansur [1], Filzah Hazirah Jaffar [1], Roziana Kamaludin [1], Mohd Hafiz Dzarfan Othman [1,*], Tutuk Djoko Kusworo [2] and Keng Yinn Wong [3]

[1] Advanced Membrane Technology Research Centre (AMTEC), Faculty of Chemical and Energy Engineering, Universiti Teknologi Malaysia, Johor Bahru 81310, Johor, Malaysia; muhammad.khalis@graduate.utm.my (K.S.); sitihawa1996@graduate.utm.my (S.H.M.N.); sumarnimansur90@gmail.com (S.M.); fhazirah5@graduate.utm.my (F.H.J.); roziana.kamaludin@utm.my (R.K.)

[2] Department of Chemical Engineering, Faculty of Engineering, Diponegoro University, Semarang 50275, Indonesia; tdkusworo@che.undip.ac.id

[3] Faculty of Mechanical Engineering, Universiti Teknologi Malaysia, Johor Bahru 81310, Johor, Malaysia; kengyinnwong@utm.my

* Correspondence: hafiz@petroleum.utm.my

**Abstract:** Bisphenol A (BPA) is an endocrine-disrupting chemical (EDC) that can cause adverse effects on human health. The incorporation of materials as visible light photocatalysts and its energy storage capability allow for the photodegradation of BPA, especially in the absence of a light source. To date, there have been no significant studies regarding energy storage in membrane technology, with only a focus on the suspension form. Hence, this study was conducted to degrade the pollutant through a co-extrusion process using a mixture of copper (II) oxide and tungsten oxide as the photocatalyst and energy storage materials, respectively. Both materials were embedded into polyvinylidene (PVDF) membranes to produce a $Cu_2O/WO_3/PVDF$ dual-layer hollow fiber (DLHF) membrane. The outer dope extrusion flow rate was set at 3 mL/min, 6 mL/min, and 9 mL/min with photocatalyst:polymer ratios of 0.3, 0.50, and 0.7 $Cu_2O/WO_3/PVDF$, respectively. The performance of the membranes for each ratio was evaluated using 2 ppm of BPA with visible light irradiation. The results showed that each membrane's outer and inner layers featured finger-like void structures, while the intermediate part had a sponge-like structure. The membrane with the photocatalyst:polymer ratio of 0.5 was hydrophilic and had a high porosity of 54.97%, resulting in a high flow of 510 L/m$^2$h. Under visible light irradiation, a 0.5 $Cu_2O$/PVDF DLHF membrane with a 6-mL/min outer dope flow rate was able to remove 97.82% of 2-ppm BPA without copper leaching into the water sample. Under dark conditions, the DLHF sample showed the capability of energy storage performance and could drive certain degradation after lighting off up to 70.73% of 2-ppm BPA. The photocatalytic DLHF membrane with the ratio of 0.5 was the most optimal due to its potential morphology and ability to degrade a large amount of BPA. It is important to emphasize that usage of materials with the capability for energy storage can provide a significant contribution toward more practical membranes, so photodegradation can occur even in dark conditions.

**Keywords:** bisphenol A; photocatalytic activity; visible light; dual-layer hollow fiber membrane; energy storage

## 1. Introduction

The development of the global industrial sector has led to an increase in the use of bisphenol A (BPA) in plastic-based products. Despite its reputation as one of the most harmful pollutants, BPA is necessary in the industrial production of epoxy resins and polycarbonate polymers. BPA is a white crystalline substance (powder or flake) with a slight phenolic odor that can be found in the environment [1]. Food containers, plastic

bottles, toys, water pipes, medical equipment, dental products, electronic devices, compact discs, thermal paper, vehicle parts, and adhesives have all been found to contain BPA [1]. Unfortunately, regardless of whether the bottles have been used or not, BPA can be released from polymer bottles at a rate of 0.20–0.79 ng/h at room temperature, and the rate of BPA release in boiling water is almost 55 times higher than that at room temperature [2]. Consequently, the discharge of BPA into the environment from plastic items has become a significant issue for health. Further technology needs to be designed to remove BPA from water for environmental protection.

Recent technologies that have been used to remove BPA from water are membrane separation, biological degradation, and chemical oxidation. Because conventional water treatment plants are ineffective at removing BPA and other new pollutants, there is growing demand for water treatment technology to ensure safe drinking water. It was reported that membrane usage for BPA removal may be divided into two categories, which include reverse osmosis (RO) and nanofiltration (NF), that use a sieving mechanism to exclude BPA molecules. The affinity (AF) membrane is the other type, which uses an adsorption process to remove BPA molecules. BPA is resistant to conventional techniques, such as coagulation-flocculation, sedimentation, adsorption, or membrane filtration for effective removal and degradation [3–6]. In contrast, advanced oxidation processes (AOPs) offer state-of-art methods for enhancing the removal of BPA through degradation techniques, such as ozonation, fenton, ultrasound irradiation, electro-fenton, and photoelectrocatalysis [7–11]. Homogeneous AOPs employing various transition metals, including Fe, Co, Cu, and Mn, have been widely used [12], but these methods have limitations, such as high chemical consumption, generation of a large amount of precipitate, and pH regulation challenges, that make them less efficient as photocatalysts. Additionally, the presence of soluble metal ions in the treated effluent and their potential environmental impact are serious concerns, necessitating their recovery [13,14]. Heterogeneous photocatalysis has higher mineralization efficiency and can break down BPA into non-toxic products, such as $CO_2$ and $H_2O$.

In 1911, Dr. Alexander Eibner, who was born in Germany and is recognized for being an inventor, made the very first discovery of the concept of a photocatalyst [15]. Under the influence of light, he discovered that zinc oxide had the ability to reduce the dark blue pigment known as Prussian blue [16]. In comparison to other methods of wastewater treatment, the photocatalysis process not only has a low cost, but it is also reusable, eco-friendly, and capable of degrading totally [17]. Depending on the energy of the photocatalyst's bandgap, this reaction can occur when the photocatalyst is exposed to either visible light or ultraviolet light. The process of photocatalysis is affected by a specific number of parameters, including the intensity of the light, the quantity of the catalyst, the temperature, the structure, the size, the pH, the surface area, and the concentration of the pollutants. This method has seen application in the removal of organic pollutants, most notably in wastewater treatment, as well as in the creation of hydrogen, antibacterial processes, and purifying of the air [18]. It eliminated contaminants in water, such as organic chemicals, pharmaceutical medicines, textile dyes, and EDCs [19–21].

To date, there is a significant problem by which light energy is required to perform photocatalytic activity. To ensure a continuous process of photodegradation of BPA, the introduction of an energy storage material would be a potential application to the membrane. Hence, the ability of tungsten trioxide ($WO_3$) semiconductors to store energy make them suitable for energy storage applications in water treatment [22]. $WO_3$ also has a suitable band gap for visible light absorption [23]. By coupling it with a photocatalyst, the energy stored can be used after dark to make it more practical.

Recently, the fabrication of photocatalytic DLHF has been in the spotlight in the research field. In the meantime, many applications of polyvinylidene fluoride (PVDF) in the fabrication of photocatalytic DLHF have been attempted, made up of an inner layer made of PVDF and a composite outer layer made of PVDF/photocatalyst. Based on different studies, PVDF seems to yield great advantages to the fabricated membrane

compared to other polymeric materials because of it open structure, thermal stability, mechanical strength, relative chemical inertness, and large availability of pore sites [24].

Previous studies by Mohamed et al. [22] have shown that visible light photocatalysts can effectively degrade bisphenol A (BPA), and photocatalysts for visible light have been developed, including copper (I) oxide ($Cu_2O$) embedded in polyvinylidene fluoride (PVDF) DLHF membranes, to degrade BPA. These membranes exhibited high hydrophilicity and high porosity of up to 63.13%, resulting in high flux of up to 13,891 $L/m^2h$, and removal efficiency of 75% for 10-ppm BPA under visible light irradiation was also attained by optimizing the $Cu_2O$/PVDF DLHF membrane with a 0.50 $Cu_2O$/PVDF ratio. However, there is still a need to further improve the efficiency and practicality of membrane technology for BPA degradation. To address this need, this study proposes the incorporation of an energy storage capability into membrane technology for enhanced BPA degradation. To date, no significant amount of research on energy storage in membrane technology has been conducted. Additionally, previous studies of photocatalytic energy storage have solely focused on the suspension form, neglecting the potential benefits of membrane technology. The novel contribution of this study is the application of energy storage materials integrated into membrane technology for BPA photodegradation. Moreover, this study builds upon the previous research by incorporating a combination of $Cu_2O$ and $WO_3$ as the photocatalyst, in contrast to a previous study that only employed $Cu_2O$. The suggested $Cu_2O$/$WO_3$/PVDF DLHF membrane has the potential to achieve photodegradation with a low-intensity light source, which is a significant advancement toward more efficient and practical membrane technology. Thus, this study is an improved version of the previous study, and it aims to address the limitations and expand the knowledge on the incorporation of energy storage materials into PVDF membranes for enhanced BPA degradation.

In this project, a compact hybrid photocatalytic membrane system, which combines advanced membrane fabrication processes and a catalyst designed for activity under visible-light and dark conditions, was developed and assembled for efficiently treating BPA, which is commonly present within water systems. The main objectives involved in this project included producing high performance photocatalytic DLHF that is actively functional under visible light and dark conditions and, last, to evaluate the BPA degradation efficiency using the developed photocatalytic membrane systems.

## 2. Results and Discussion

### 2.1. Morphological Analysis and EDX Mapping

The effects on length can be explained by examining the hydrophilicity and viscosity of the dope solution, as both factors could affect the solvent/non-solvent exchange rate during the phase inversion process. Therefore, it can be seen that the finger-like length increased with increase in the photocatalyst: polymer ratio up to 0.7 (Figure 1). In contrast, the finger-like length at the inner layer became shorter as the fraction increased to 0.7. This outcome might be due to the hydrophilicity properties of $WO_3$/$Cu_2O$ in the outer layer, which would lead to the attraction of a large amount of non-solvent (water) flow from the bore fluid (lumen) toward the outer layer via the inner surface of the nascent fiber. This process resulted in the enlargement of the finger-like void volume from the inner surface when more $WO_3$/$Cu_2O$ was added to the outer layer. It can be seen that the finger-like length in the inner layer increased when the $WO_3$/$Cu_2O$/PVDF ratio was increased.

The SEM photos demonstrate the DLHF membranes' outer surfaces at various ratios. Figure 2 shows the outer surface morphology of the $Cu_2O$/$WO_3$/PVDF membrane with different ratios. Meanwhile, increasing the ratios of $Cu_2O$/$WO_3$/PVDF in the dope solution was expected to increase the number of $Cu_2O$/$WO_3$/PVDF particles deposited on the membrane surface. $Cu_2O$/$WO_3$/PVDF particles agglomerated more significantly as the ratio of $Cu_2O$/$WO_3$/PVDF increased, as illustrated in Figure 2. The greater surface tension between the solvent of the dope solution and $Cu_2O$/$WO_3$/PVDF nanoparticles may have caused agglomeration.

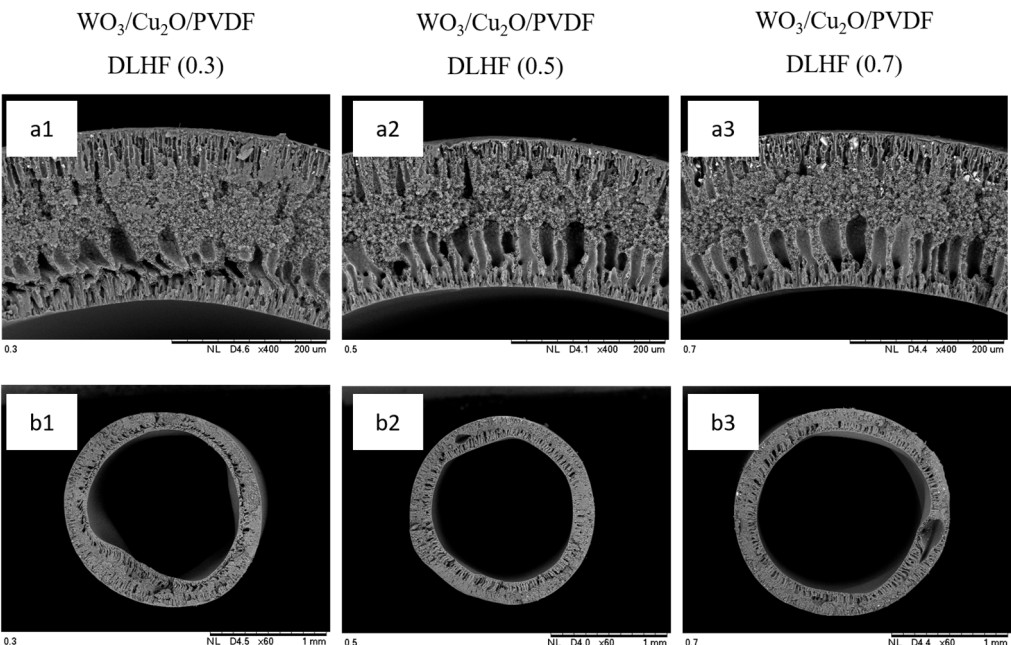

**Figure 1.** SEM cross-sectional, morphological analysis of DLHF membranes of different $WO_3/Cu_2O/PVDF$ loadings of 0.3, 0.5, and 0.7. Figures (**a1**–**a3**) and (**b1**–**b3**) represent cross-sectional magnifications of 400 μm and 60 mm, respectively.

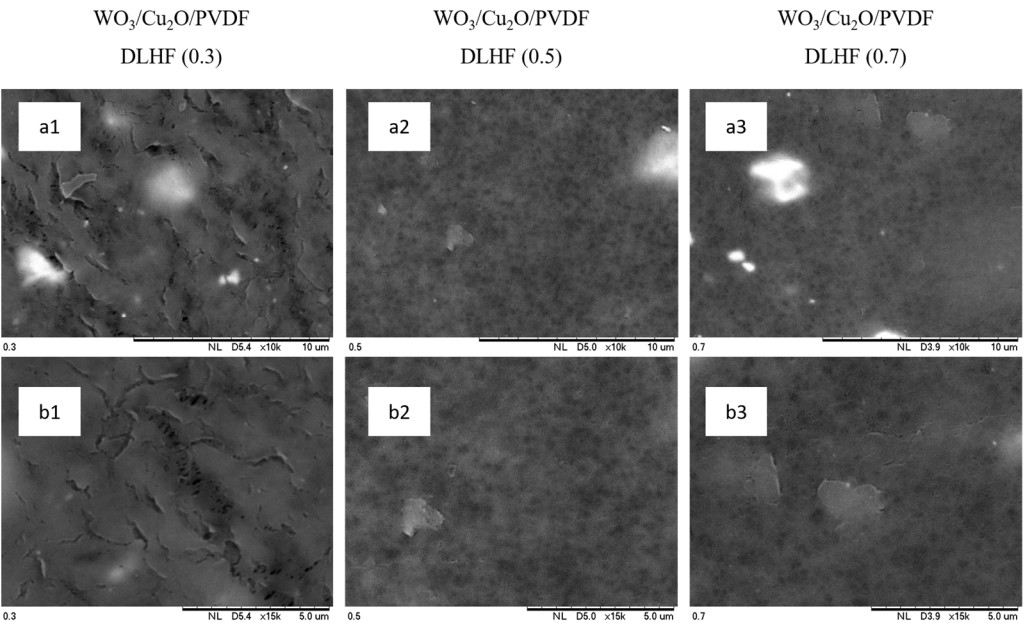

**Figure 2.** SEM surface morphologies of $Cu_2O/WO_3/PVDF$ DLHF with different $Cu_2O/WO_3$ loadings (Magnification (**a1**–**a3**): ×10 k, (**b1**–**b3**): ×15 k).

Less photocatalyst particle diffusion into the inner layer DLHF would be beneficial to the DLHF membrane because the thin outer layer would allow for a large amount of penetration of the light source, resulting in a larger active reaction site for photocatalytic activity. EDX analysis images, as shown in Figure 3, confirmed the homogeneous distribution of photocatalyst (represented by the Cu element) on the membranes' outer surface. As shown in Figure 3, the EDX image reveals the presence of the photocatalyst on the outer membrane surface. Furthermore, the presence of photocatalyst on the outer membrane surface had a significant impact on the membrane photocatalytic activity when exposed

to visible light. As a result, even with inadequate indoor lighting, the photodegradation process appears promising.

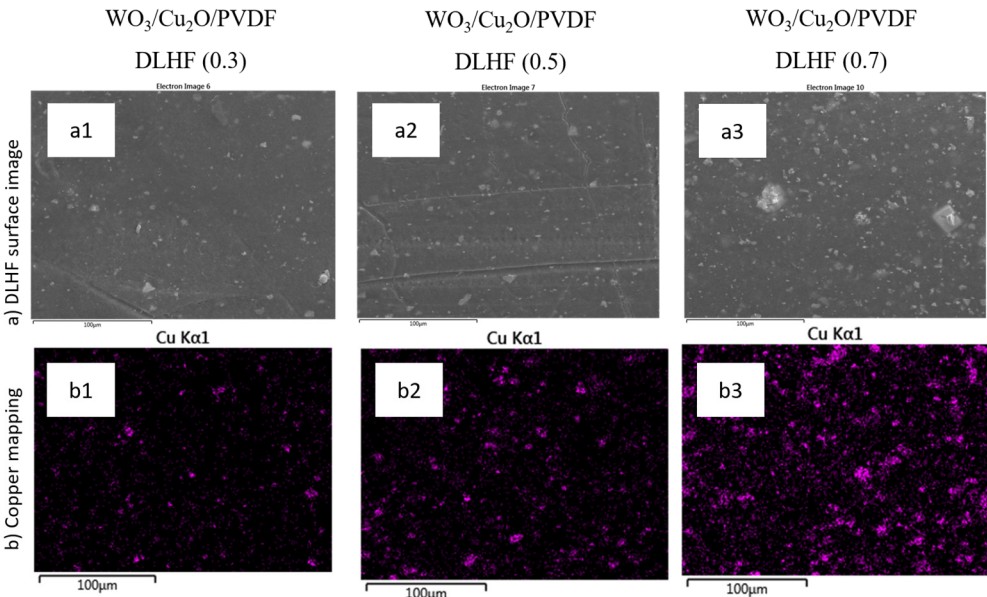

**Figure 3.** EDX mapping surface image of $Cu_2O/WO_3/PVDF$ DLHF membranes with different $Cu_2O/WO_3$ loadings of 0.3, 0.5, and 0.7 (Magnification: **a1–b3**: ×10 k).

The outer layer in this membrane structure primarily functions as a degradation area for the contaminant by photocatalytic activity. The inner layer acts as a separation barrier. As shown in Figure 4, the EDX images of cross section of $Cu_2O/WO_3/PVDF$ DLHF was determined. The majority of photocatalysts can absorb organic contaminants prior to photodegradation. As a result, during the performance evaluation, the fabricated $Cu_2O/WO_3/PVDF$ DLHF adsorption capability was tested using a BPA adsorption test in aqueous solution. The organic pollutant was caught by active sites of the photocatalyst on the outer layer of the membrane surface, and only $H_2O$ molecules and other types of pollutants were allowed to pass through it. The degrading occurred on the outer membrane layer after a specific amount of time, and the product of the photocatalytic degradation process penetrated the inner membrane layer. Then, another organic pollutant adsorption process occurred, and the cycle repeated. As a result, the main mechanism of photocatalytic DLHF photocatalytic degradation, with the outer layer serving as the degradation site and the inner layer serving as the separation layer.

The photocatalyst has been shown to change the affinity of the membrane for water. The results of the contact angle and the contact angles of all DLHF membranes are summarized in Table 1. All DLHF membranes are hydrophilic, according to the research. The hydrophilic nature of $Cu_2O/WO_3$ particles present in the outer layer of DLHF membranes successfully improved the membrane's affinity for water. The contact angles for $Cu_2O/WO_3$ DLHF (0.2), (0.5), and (0.7) were 70.26°, 70.95°, and 72.98°, respectively, as shown in this table. Because more $Cu_2O/WO_3$ particles are integrated into the outer membrane layer, $Cu_2O/WO_3$ DLHF (0.7) is less hydrophilic than $Cu_2O/WO_3$ DLHF (0.2) and (0.5) since the hydrophilic property of the membrane is determined by the distribution of the hydrophilic material over the membrane surface, so the hydrophilicity improved as the distribution of the hydrophilic nature increased. Increased membrane hydrophilic characteristics are beneficial to water-related separation processes, as well as photocatalytic reactions; therefore, contaminant degradation and separation appear to be promisingly efficient. The porosity of the $Cu_2O/WO_3/PVDF$ DLHF membrane is shown in Table 1. The increased membrane pore size is responsible for the increased membrane porosity [3]. Microfiltration and ultrafiltration membranes made of PVDF have pore diameters ranging

from 20 to 450 nm [3]. The porosity of the PVDF membrane is less than 60%, and this value is sufficient for the mmbrane to act as an ultrafiltration membrane, which requires 30–60% porosity. The study confirms that increasing the pore size increases the membrane porosity and that pore size selectivity determines the membrane's application.

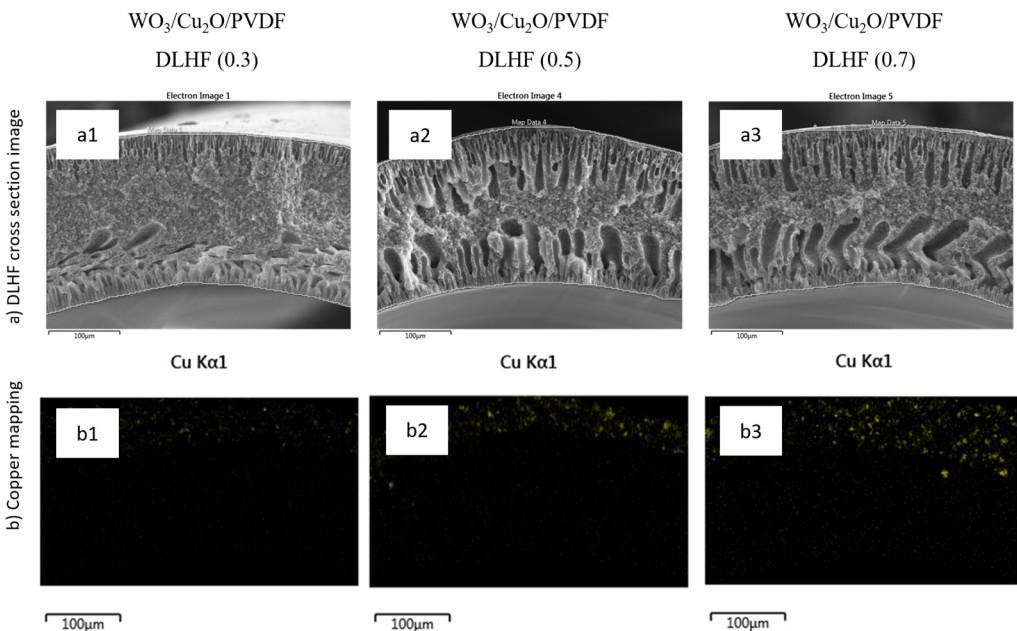

**Figure 4.** EDX mapping cross-section image of $Cu_2O/WO_3/PVDF$ DLHF membranes with different ratios of 0.3, 0.5, and 0.7 (Magnification (**a1–b3**): 100 μm).

**Table 1.** Performance of the $Cu_2O/WO_3/PVDF$ DLHF membrane.

| Membrane Ratio | Contact Angle (°) | Porosity (%) | Mean Pore Size | Water Flux (L/m² h) |
|---|---|---|---|---|
| 0.3 | 70.26° ± 2.42 | 50.83% ± 1.92 | 196.5 nm ± 7.18 | 108.86 ± 19.89 |
| 0.5 | 70.95° ± 0.98 | 54.97% ± 3.85 | 278.0 nm ± 9.28 | 510.00 ± 67.14 |
| 0.7 | 72.98° ± 0.95 | 50.10% ± 1.50 | 67.48 nm ± 4.72 | 295.07 ± 9.18 |

### 2.2. Functional Group Analysis

The particular absorption of infrared light in vibrational modes of Fourier transform infrared (FTIR) spectroscopy is a helpful approach for identifying chemical bond functional groups. At ambient temperatures, the FTIR spectra of the manufactured membrane $Cu_2O/WO_3/PVDF$ DLHF (0.5) samples are shown in Figure 5a. The FTIR spectra were obtained in the region of 400–4000 $cm^{-1}$. $Cu_2O$ absorbs infrared photons with the greatest absorptions at 1130, 798, and 621 $cm^{-1}$, which correspond to Cu-O stretching vibrations in $Cu_2O$. Absorption peaks were seen at 797 and 625 $cm^{-1}$ here. The findings established the presence of $Cu_2O$ [25,26].

The W-OH vibration is represented by the peak at 1622 $cm^{-1}$ in the $WO_3$ spectrum. Strong absorption peaks may be found at 893, 961, 814, and 700 $cm^{-1}$ for the W-O-W bond, whereas an absorption peak can be found at 962 $cm^{-1}$ for the W=O and W-O bonds. In general, the absorption peaks listed above can overlap and form a broad peak in the range of 1270 to 445 $cm^{-1}$ [27].

The crystalline peaks of $Cu_2O/WO_3/PVDF$ DLHF (0.5) membranes were 2θ = 36.46°, 42.30°, 44.52°, 61.34°, and 73.70° which were analogous to the peaks of the PVDF membrane's $WO_3$ and $Cu_2O$ particles (Figure 5). This finding indicates that interactions between PVDF polymer and $WO_3$ and $Cu_2O$ particles influenced the PVDF crystal structure (transition from a to ß phase) in the neat PVDF powder, exhibiting peaks at 2θ = 18.69° and 2θ = 20.11°, which were representative of a-polymorph (Figure 5), which was close to

an earlier published study [24]. The ratio 0.5 was chosen to represent the DLHF sample because it was expected that there would not be much difference seen for other DLHF samples.

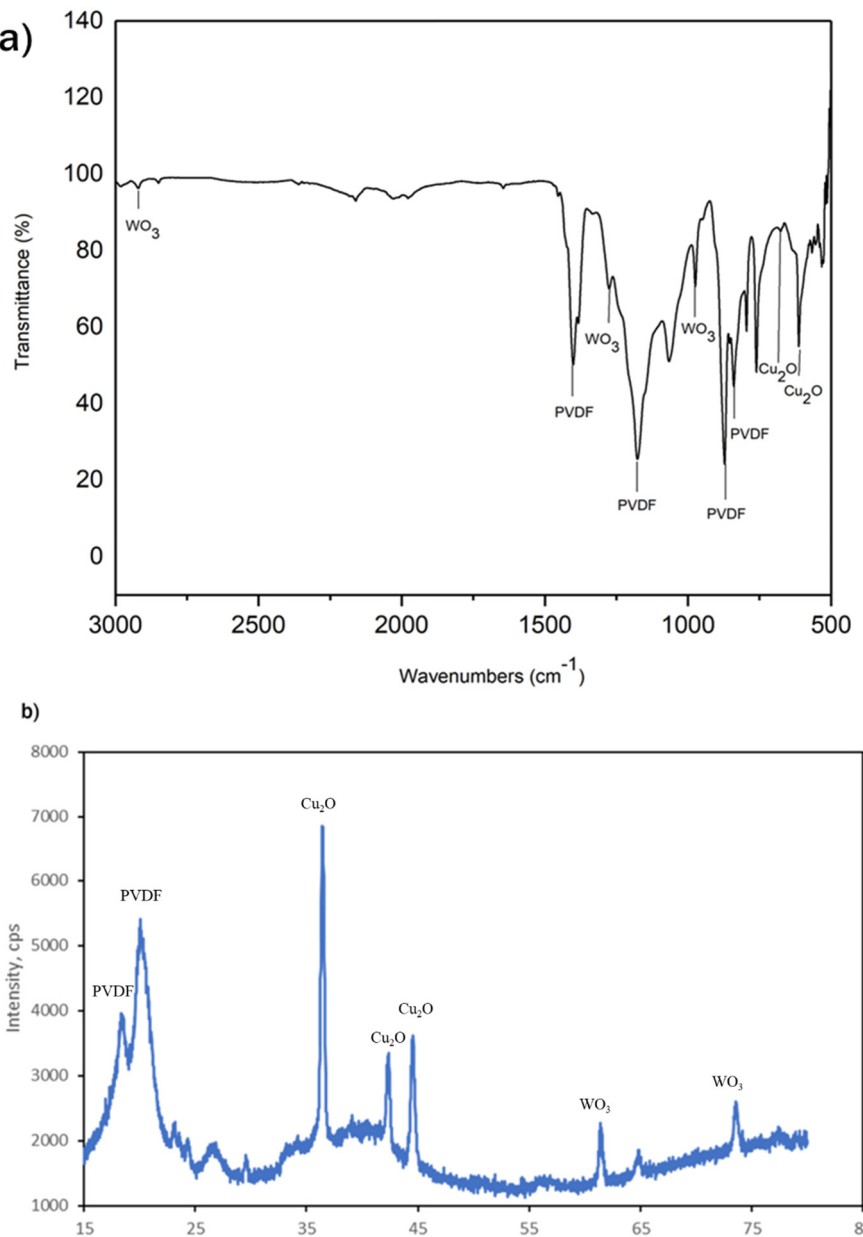

**Figure 5.** (**a**) FTIR pattern and (**b**) XRD pattern of $Cu_2O/WO_3/PVDF$ DLHF.

### 2.3. Photocatalytic Activity Performance

The membrane was subjected to adsorption-desorption equilibrium testing, which required 120 min of adsorption-desorption equilibrium before light irradiation could begin. In addition, the photocatalytic performance of the outer dope at a flow rate of 6 mL/min for all three membrane ratios was investigated for decomposing 2 ppm of BPA, as shown in Figure 6. The 0.5 ratio $Cu_2O/WO_3/PVDF$ DLHF membrane had superior photocatalytic activity in degrading BPA compared to the 0.3 and 0.75 ratios. A 0.5 ratio $Cu_2O/WO_3/PVDF$ membrane degraded 97% of 2 ppm BPA in 360 min under visible light.

**Figure 6.** BPA mineralization pathway.

Hydroxyl radicals (OH$^\bullet$) were suggested to be responsible for heterogenous BPA degradation. The $Cu_2O/WO_3$ DLHF was irradiated by visible light, resulting in generation of electron e$^-$ in the CB, leaving a hole h$^+$ in the VB, which then reacted with the BPA absorbed on the catalyst surface. BPA reacts with the hydroxyl radicals generated and attacks different carbon atoms in the BPA molecules, leading to formation of hydroxylated intermediates. Further hydroxylation of hydroxylated intermediate can result in the formation of multi-hydroxylated intermediates. The hydroxylated and multi-hydroxylated intermediates can undergo further degradation through mineralization to $CO_2$ and $H_2O$. Figure 6 shows the complete mineralization pathway of BPA.

As shown in Figure 7, over 360 min of visible light irradiation, the $Cu_2O$ single layer hollow fiber degraded BPA faster than PVDF. The efficiency of the $Cu_2O$ single layer hollow fiber was 84.38%, compared to that of the PVDF single layer hollow fiber at only 24.32%. Meanwhile, the 0.3 ratio $Cu_2O/WO_3$/PVDF DLHF membrane decomposed 93.94% of 2-ppm BPA, compared to 94.44% for the 0.7 ratio $Cu_2O/WO_3$/PVDF DLHF membrane. The value of the reaction rate constants for decomposition of BPA (k) of 0.3 $Cu_2O/WO_3$/PVDF DLHF, 0.5 $Cu_2O/WO_3$/PVDF DLHF, and 0.7 $Cu_2O/WO_3$/PVDF DLHF were $7.8 \times 10^{-3}$ min$^{-1}$, $9.10 \times 10^{-3}$ min$^{-1}$, and $7.7 \times 10^{-3}$ min$^{-1}$ respectively. The 0.3 ratio $Cu_2O/WO_3$/PVDF DLHF membrane had the lowest performance compared to the other membranes due to photocatalyst aggregation on the outer layer. The total surface area was influenced by the agglomeration of photocatalyst; hence, the exposed

surface area toward the light source was reduced when higher agglomeration of photocatalysts covered on outer layer [28]. The membrane porosity and the overall active site surface area available for photocatalytic degradation were reduced when photocatalysts were clumped together [29]. The process also increased surface roughness and clogged pores, resulting in a membrane with limited water flux. The photocatalytic activity of the $0.70 \ Cu_2O/WO_3/PVDF$ DLHF membrane was low due to the high ratio of $Cu_2O$ and $WO_3$ to PVDF. The $0.7 \ Cu_2O/WO_3/PVDF$ DLHF membrane has better performance than the $0.3 \ Cu_2O/PVDF$ DLHF membrane, but it was not as good as that of the $0.50 \ Cu_2O/PVDF$ DLHF membrane. In terms of degrading 2-ppm BPA, the $0.50 \ Cu_2O/WO_3/PVDF$ DLHF membrane performed the best. After 360 min of visible light irradiation, the N-doped $TiO_2$ DLHF membrane removed 81.6% of BPA at a starting concentration of 5 ppm, compared to the results of Kamaludin et al. [3]. Meanwhile, the ratio $0.50 \ Cu_2O/WO_3/PVDF$ DLHF membrane removed 97.82% of BPA, suggesting that the $0.50 \ Cu_2O/WO_3/PVDF$ membrane has high removal efficiency. With a uniform coating of $Cu_2O$ and $WO_3$ (at a configuration ratio of 0.5) on the membrane's surface, the photocatalytic activity can be maximized by exposing as many active sites as possible to the light source. However, having smaller particle sizes might increase the specific surface area, in turn leading to greater exposure of active sites, and the possibility of agglomeration increases with decreasing particle sizes, which might result in a reduction in efficiency. An optimal balance between particle size and distribution uniformity was achieved using a membrane surface structure of $Cu_2O/WO_3/PVDF$ with a ratio of 0.5. Compared to other configurations, the 0.5 ratio configuration provided homogeneous distribution, minimized agglomeration, and increased active site exposure, resulting in superior photocatalytic efficacy in degradation of BPA.

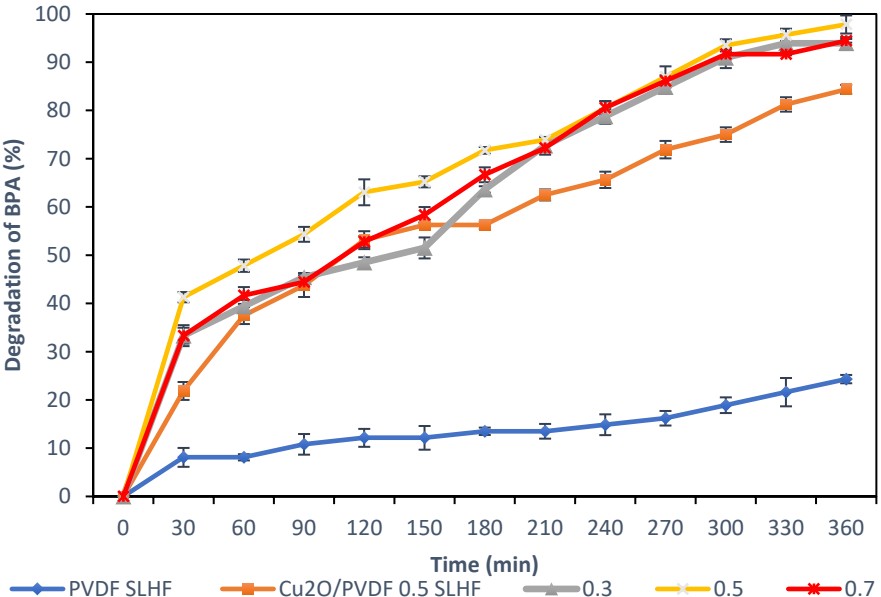

**Figure 7.** Degradation of BPA by photocatalytic performance under visible light (*n* = 3). *n* = no of times experiment repeated.

### 2.4. Energy Storage Performance

The degradation efficiencies of BPA over $Cu_2O$ and $WO_3$–$Cu_2O$ composites in the dark, after 3 h of visible light irradiation and leaving it submerged in 2-ppm BPA solution for 150 min in the dark in BPA solution, are shown in Figure 8. The pure $Cu_2O$ experiment revealed very low photodegradation of BPA, whereas $Cu_2O$ composites showed the highest degradation. The membrane was ratio 0.5 was chosen since it already was the most efficient in photocatalytic performance among the others. The energy storage processes of the $WO_3$ composites are graphically represented in the equation below. Single $Cu_2O$ exhibited no energy storage capacity, but single $WO_3$ had a sufficient energy storage capacity. In

comparison, $Cu_2O/WO_3$ samples had a substantially higher energy storage capacity than $Cu_2O$ alone, indicating the hybrid $Cu_2O/WO_3$ samples' energy storage capabilities. The main reaction of energy storage is divided into three stages: creation of photo-electrons, storage of photo-excited electrons, and release of storage electrons [29].

$$WO_3 + xH^+ + xe^- \leftrightarrow HWO_3 \tag{1}$$

$$Cu_2O + \text{visible light} \rightarrow Cu_2O\ (e_{CB-} + h_{VB'+}) \tag{2}$$

$$Cu_2O\ (h_{VB+}) + H_2O \rightarrow Cu_2O + H^+ + HO \tag{3}$$

$$Cu_2O\ (h_{VB+}) + OH^- \rightarrow Cu_2O + HO \tag{4}$$

$$Cu_2O\ (e_{CB-}) + O_2 \rightarrow Cu_2O + O_2^- \tag{5}$$

$$O_2^{-\bullet} + H^+ \rightarrow HO_2 \tag{6}$$

$$HO_2^\bullet \rightarrow O_2 + H_2O_2 \tag{7}$$

$$H_2O_2 + O_2^{-\bullet} \rightarrow HO^\bullet + OH^- + O_2 \tag{8}$$

$$HO^\bullet + BPA \rightarrow \text{products} \tag{9}$$

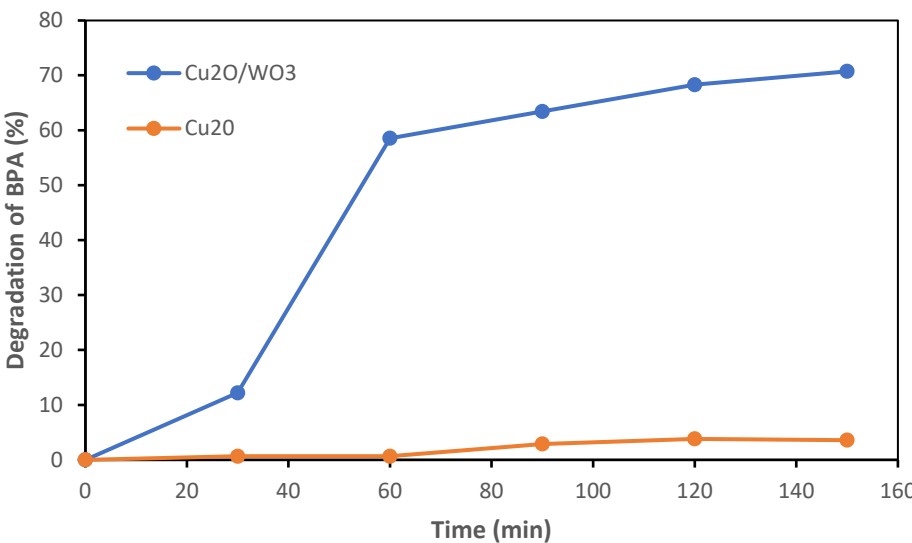

**Figure 8.** Degradation of BPA under dark conditions for 150 min using $Cu_2O/WO_3/PVDF$ and $Cu_2O/PVDF$ DLHF membranes.

The mechanism of photocatalysis and energy storing is a complex process, as shown in Equation (2). The process starts once a light strikes a semiconductor's surface in the form of a photon. If the light ray is equal to or greater than the energy bandgap, the electrons on the valence band become excited and strike the semiconductor's conduction band. As the electron moves, it leaves holes in the valence band. The holes oxidize the $H_2O$, act as donor molecules, react with it, and form hydroxyl radicals. The hydroxyl radicals are responsible for pollutant degradation since they have the strong power to oxidize. The electrons on the conduction band interact with oxygen and undergo reduction from superoxide ions. Overall, a redox reaction occurs in photocatalysis and is induced by electrons [18]. Furthermore, the charge generated by the photocatalytic reaction is stored when electrons produced by the reaction are trapped by the intercalation of ions. These protons ($H^+$)

are intercalated into the $WO_3$ lattice, forming positively charged hydrogen ions ($H^+$) that remain in interstitial sites. This process, resulting in positively charged $WO_3$, helps to build an electrode that can store the charge produced by the absorption of light. This removal of protons and electrons contributes to the release of the stored charge in dark conditions, making it available for use by the photocatalyst [30,31]. The stored electrons are released after the light is switched off, and they can react much like photo-electrons. Figure 8 below shows two samples of membranes, one consisting of $Cu_2O$ and $WO_3$ particles and the other containing only $Cu_2O$. The figure shows that degradation occurs in the existence of $WO_3$ in the $Cu_2O/WO_3/PVDF$ DLHF membrane, which contributes to 70.73% of degradation of 2-ppm BPA under dark conditions. Meanwhile, the $Cu_2O/PVDF$ DLHF membrane only contributes to 3.57% of the degradation of 2-ppm BPA.

By contrasting the outcomes of the current study with those of the previous one, $Cu_2O$ was investigated in both membrane configurations for the removal of BPA. Whereas Mohamed Noor et al.'s [22] earlier research looked at incorporating $Cu_2O$ into PVDF DLHF membranes, the present study instead concentrated on $Cu_2O/WO_3$ DLHF membranes. The authors showed that a membrane design of 0.50 $Cu_2O/PVDF$ DLHF was the most effective at removing 75% of 10 ppm BPA over 360 min when exposed to visible light. This research confirmed previous findings that the optimal membrane design for photocatalysis was 0.50 $Cu_2O/WO_3$ DLHF, which degraded 90% of BPA in the same amount of time as in the previous study. This membrane also degraded 70.73% of BPA in dark conditions. This study demonstrated that $Cu_2O/WO_3$ membranes may provide even better results than those shown in the previous study, which focused on employing $Cu_2O$ in PVDF membranes. In addition, the ability of $Cu_2O/WO_3$ DLHF membranes to achieve high levels of BPA degradation, even in poor situations, shows the potential of these materials for interesting applications.

Synthetic polymeric membranes, such as PVDF membranes, can undergo photodegradation under certain conditions [32]. When a photocatalyst is exposed to light, reactive oxygen species (ROS) are produced, creating a highly oxidizing environment. As a result of their interactions with the polymeric membrane surface, these ROS can cause chain scission, crosslinking, and surface erosion due to their high oxidizing power [33]. The membrane's performance and stability may decrease as a result of degradation, which reduces its mechanical strength, permeability, or selectivity. The incident light consists of three ranges of electromagnetic spectra (visible, ultraviolet, and infrared), and each causes polymer degradation, although UV radiation has the most energy, despite it only contributing 8% of the total incident [34]. The mechanical strength and thermal stability of irradiated PVDF-based membrane were reported to decrease with increasing UV exposure time. It was stated in a previous study that the cracks and fractures were detected on PVDF-based membrane surfaces when the membrane was exposed directly to UV light for up to 60 h. Additives need to be included in polymers before they can be utilized in commercial applications. Some examples of commercial additives are UV and heat stabilizers, impact and thermal modifiers, flame retardants, blowing reagents, and smoke suppressors [35]. The primary goals of this project are to create a dual-layer hollow fiber membrane with exceptional photocatalytic properties that can operate efficiently under both visible light and dark conditions. Hence, a better understanding and future studies of membrane performance could expose potential photodegradation risks, and techniques to reduce such risks need to be considered. This study established a foundation for future studies of membrane materials to investigate their possible uses.

## 3. Materials and Methods

### 3.1. Membrane Materials

The materials used to fabricate the membrane were polyvinylidene fluoride (PVDF, Kynar 760 Series-powder, Solvay Specialty Polymers France) (molecular weight: 441,000 by GPC), polythethylene glycol 6000 ((PEG, Sigma Aldrich, Burlington, MA, USA), dimethylene acetamide (DMAc, Sigma Aldrich, Burlington, MA, USA), ethanol (EtOH, Hayman,

Australia), copper (II) oxide ($Cu_2O$, Sigma Aldrich, Burlington, MA, USA), and tungsten oxide ($Cu_2O$, Sigma Aldrich, Burlington, MA, USA).

### 3.2. Fabrication of Photocatalytic DLHF Membrane

### 3.2.1. Inner Layer Dope Preparation

The preparation inner layer dope began with PEG 6000 being dissolved by DMAc in the amount required as listed in Table 2 in a Scott bottle with an overhead stirrer (IKA RW 20 digital), followed by stirring at 600 rpm for 24 h. PVDF was added to the mixture once it reached a homogeneous state. After the PVDF/PEG/DMAc dope solution became homogeneous, the solution was cooled to room temperature and stored in a dry area until the spinning day. Before the spinning day, the inner dope solution was sonicated overnight at room temperature to remove air bubbles and homogenize the dope, contributing to a smooth membrane structure.

**Table 2.** Polymer dope solutions compositions.

| Membrane Configuration | Outer Layer Composition (wt. %) | | | | Inner Layer Composition | | |
|---|---|---|---|---|---|---|---|
| DLHF membrane | PVDF | $Cu_2O$ | $WO_3$ | DMAc | PVDF | PEG 6000 | DMAc |
| 0.3 | 15 | 1.25 | 2.5 | 81.25 | 15.0 | 3.0 | 82.0 |
| 0.5 | 15 | 2.5 | 5.0 | 77.5 | 15.0 | 3.0 | 82.0 |
| 0.7 | 15 | 3.75 | 7.5 | 73.75 | 15.0 | 3.0 | 82.0 |

### 3.2.2. Outer Layer Dope Preparation

$WO_3$ and $Cu_2O$ were dissolved in a Scott bottle with an overhead stirrer (IKA RW 20 digital) and stirred for 24 h at 600 rpm by DMAc in the necessary amount, as shown in Table 2. PVDF was added to the mixture once it reached a homogeneous state. A sample PVDF/$Cu_2O$/$WO_3$ DLHF membrane could be modified to have different ratios of 0.3, 0.5, and 0.7 [3]. Before starting the spinning process, the outer dope solution was sonicated for half an hour at 180 W to ensure that the photocatalyst is evenly distributed throughout the solution and to remove any trapped air bubbles. All of the dope solution then underwent viscosity measurements using a basic 20-2 million centPoise viscometer (Cole Parmer, Model: EM-98965-40, Vernon Hills, IL, USA).

### 3.2.3. Dry/Wet Co-Spinning Technique

The visible light photocatalyst DLHF was fabricated by the technique of dry/wet co-spinning with specific conditions, the method for which has been reported elsewhere [3]. The inner and outer dope flow rates were set at 26 rpm and 1–4 rpm, respectively. Distilled water was used as a bore fluid at 8 rpm. The air gap between the spinning rate and coagulation bath was set to 100 mm. The take-up speed for the rotating drum was set up to 5 rpm. After ultrasonication, the dope solutions were ready to be transferred into a stainless-steel dope reservoir syringe pump and then extruded via a triple orifice spinneret to form a DLHF membrane. The phase inversion process began once the dope solution entered the coagulation bath, which was composed of a non-solvent for the purpose of solidification. After collecting as-spun hollow fibers from the collector, they were placed in a tank of deionized water for 24 h to remove any remaining diluent. The as-spun hollow fiber membrane was then post-treated for 1 h in 50% ethyl alcohol. To avoid membrane shrinking, the as-spun hollow fibers were submerged in 100% ethyl alcohol for 1 additional h. The PVDF hollow fiber membrane was then air dried for 24 h at room temperature.

### 3.2.4. Post-Treatment

The fabricated membranes were soaked into a water bath to remove remaining solvent for 24 h. Then, the membranes were placed in bundles of 50 pieces 30 cm in length and cut. A few bundles of membrane were immersed in ethanol:water, 50:50 wt. % for 60 min and treated again with 100% ethanol for another 60 min. This treatment enhances

the membrane's wet ability, and the pores fall in, thus decreasing the resistance of mass transfer [36]. The final step for post-treatment was to hang the membranes and dry them at room temperature for 3 days and then to keep in zipped plastic and store them until further analysis.

### 3.3. Characterization Methods

3.3.1. Field Emission Scanning Electron Microscope (FESEM)

Field emission scanning electron microscopy (FESEM) is a method for examining the DLHF membrane's structural properties. The membranes were analyzed using a FESEM (Model: SU2080, Hitachi, Tokyo, Japan) with a 2.0-kV accelerating voltage. To maintain the structure of the cross-section, the samples of membrane were immersed in liquid nitrogen for 10 s until the samples fractured and were cut into halves. The samples were then placed on the metal holder and sputter coated with gold under a vacuum for about 3 min to discharge the samples.

3.3.2. X-ray Diffraction (XRD)

The crystallinity and phase identification of photocatalytic $Cu_2O/WO_3/PVDF$ DLHF were investigated using XRD (Model: D5000, SIEMENS, Munich, Germany). The test was performed at 40 kV and 30 mA. Furthermore, it used CUK-ß radiation with a wavelength of 0.15418 nm, an angular incidence of $2\theta = 20$–$800$, and a scan step speed of $1°/min$.

3.3.3. Energy Dispersive X-ray (EDX)

Energy dispersive X-ray (EDX) is a technique for detecting photocatalyst dispersion on the membrane surface. After placing the DLHF membranes in the metal instrument holder (EDX; Model: X-MaxN 51-XMXIOII, Oxford Instrument, Abingdon, Oxfordshire, United Kingdom), the samples were immersed in liquid nitrogen for 60 s. The samples were fractured into shorter pieces to make the cross-sectional particle dispersion analysis easier. Under a vacuum, the samples were sputter coated with gold for 3 min. Under multiple magnifications, the surface and cross-section of the DLHF membrane were examined, and then the samples were scanned with EDX.

3.3.4. Fourier-Transform Infrared Spectroscopy (FTIR)

Fourier-transform infrared spectroscopy (FTIR) is used to detect functional groups and chemical bonds after a sample absorbs the infrared spectrum. A Perkin Elmer FTIR attenuated total reflection (ATR) spectrophotometer with a diamond ATR sample attachment was used to provide a non-destructive means for DLHF membrane measurement. The samples were clamped with the ATR diamond. Then, the pressure was applied to ensure optimal contact of the surfaces with each other. The samples were scanned with a wavelength spectrum ranging from 650 to 4000 cm$^{-1}$.

3.3.5. Contact Angle

The adhesive and cohesive forces act on liquids. The contact angle of the membrane had to be measured, and it ranged from 0 to 180 degrees. The adhesive reaction occurs when a liquid meets a solid, while the cohesive reaction occurs when a liquid meets another liquid. The photocatalytic DLHF membrane was evaluated using a contact angle goniometer (Model: OCA 15EC, Dataphysic, Filderstadt, Germany) with image processing software that is packaged with the analyzer to analyze the degree of hydrophilicity. Deionized water as a contact liquid was dropped on the membrane surface at various spots. Seven readings were recorded, and the average and standard deviation were calculated.

3.3.6. Membrane Porosity

The porosity of the membrane was determined by substituting the significant values in the gravimetric method, as shown in Equation (10). First, the samples were cut into

three pieces of 7 cm each, and their weight as dry samples was observed. Next, the samples were immersed in water for 24 h, and the weight of the wet samples was observed.

$$\varepsilon\ (\%) = \frac{\frac{(W_W - W_d)}{\rho_H}}{\frac{(W_W - W_d)}{\rho_H} + \frac{W_d}{\rho_c}} \tag{10}$$

where $W_W$ and $W_d$ are the weights of wet and dry membrane (g), respectively. $\rho_H$ is the density of water (0.998 g/cm$^3$), and $\rho_c$ is the density of cellulose (1.5 g/cm$^3$).

### 3.3.7. Flux Rate Analysis

The flux of pure water was analyzed using a membrane module filtration apparatus. Initially, three fibers of DLHF membrane 11 cm in length were made into a bunch, which was then placed in the filtration module. The water flux in the mode of cross-flow through an outside-in configuration was measured. Distilled water was used in the flux rate analysis. In this analysis, the permeability of the membrane was obtained by applying different pressures to different samples of membranes. Pressure of 0.15 MPa to the membrane for 10 min was applied to obtain a constant flux by compacting the membrane. Then, the measurement of flux was started at 0.1 MPa, and the value was calculated according to Equations (11) and (12) below:

$$F = \frac{V}{A \times t} \tag{11}$$

$$A = \pi do L \tag{12}$$

where $F$ is equal to the membrane flux unit of L/m$^2$h, $V$ is equal to the permeation volume at time (L), $A$ is the membrane filtration area (m$^2$), $do$ is the hollow fibers' outer diameter (cm), and $L$ is equal to the hollow fibers' effective length (cm).

### 3.4. Photocatalytic Performance

#### 3.4.1. Preparation of Bisphenol-A Solution as BPA-Contaminated Water

To determine the photocatalytic activity of DLHF membrane on bisphenol-A, BPA-contaminated water was prepared. BPA was supplied by Hovid Berhad and straightforwardly used in the experiment without any purification needed. One gram of BPA powder was weighed and dissolved with deionized water: acetonitrile equal to 9:1 in a 1000-mL volumetric flask was used to obtain a 1000-ppm stock solution. One milliliter of BPA stock solution was pipetted into a 500-mL volumetric flask, and deionized water was added to the mark to obtain 2 ppm.

#### 3.4.2. Photocatalytic Performance of DLHF Membrane

A submerged photocatalytic system was built to achieve the goal of analyzing the photocatalytic performance of the Cu$_2$O/WO$_3$/PVDF DLHF membrane in the removal of BPA from wastewater. DLHF membranes of 12 were assembled in a U-shape with a diameter of approximately 20 cm. Sample membranes were placed in PVC tubes using epoxy resin (E-30CL Loctite Corporation, Rocky Hill, CT, USA) and allowed to harden at room temperature. Then, the PVC adapter was attached to the potted membranes in PVC. Since the photocatalyst activates and reacts effectively under visible light, the system includes a visible light lamp (LED Flood Light; Model: IP66, 30 W) in the middle of the water tank. Wastewater containing BPA (2 ppm in 10 L of distilled water) was fed into the feed tank, and it flowed down to membrane unit driven by gravity. The membrane module was immersed into the membrane unit, and a peristaltic pump was started to collect the BPA-contaminated water solution into the permeated tank as treated water.

### 3.4.3. Photocatalytic Performance of DLHF Membrane under Dark Conditions

To analyze the energy storage performance of the $Cu_2O/WO_3/PVDF$ DLHF membrane for BPA removal under dark conditions, the solution was first placed in the same setup as in Figure 9 for 120 min until it reached adsorption/desorption equilibrium. The solution was then irradiated under visible light for 30 min, and aliquots of 10 mL of the solution were collected at 30-min intervals until 360 min passed. The collected solutions were analyzed for BPA removal efficiency using UV-Vis spectrophotometry.

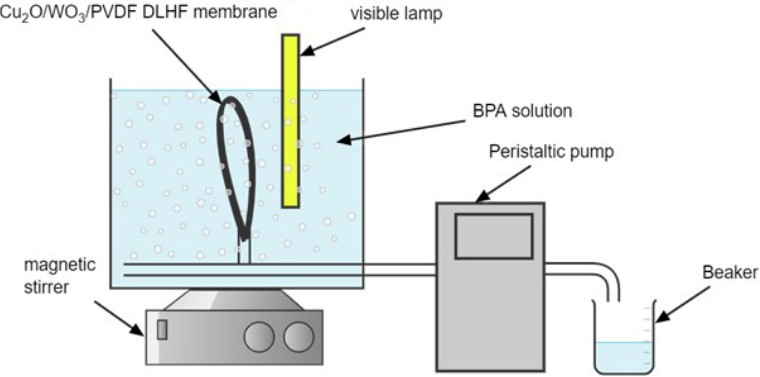

**Figure 9.** Submerged photocatalytic reactor.

### 4. Conclusions

The fabrication of a visible light-driven photocatalytic DLHF membranes has been the focus of this study. A single step co-extrusion process was used to produce $Cu_2O/WO_3/PVDF$ DLHF membranes with various $Cu_2O/WO_3/PVDF$ ratios. The fabricated DLHF membranes exhibited a characteristic DLHF shape, with finger-like voids at the outer and inner layers and a sponge-like structure in the middle. The addition of multiple ratios of $Cu_2O/WO_3/PVDF$ to the DLHF membranes had a substantial impact on the finger-like void, membrane surface roughness, hydrophilicity characteristics, and energy storage performance, with 0.5 as the best ratio. The potential ratio was observed to be 0.5, as it can degrade up to 97.82% of 2-ppm BPA solution. Additionally, the ability of semiconduction in the membrane to store energy drove greater photocatalysis, even with no presence of light. As a result, even under dark conditions, the fabricated $Cu_2O/WO_3/PVDF$ DLHF was able to undergo photodegradation of BPA by up to 70.73%. This outcome shows that immobilization of $Cu_2O/WO_3$ nanoparticles in the PVDF hollow fiber membrane matrix provides a good treatment option for the total elimination of a wide range of BPA. The best spun visible-light driven $Cu_2O/WO_3/PVDF$ DLHF was created with a sandwich-like structure and good creation of finger-like voids at the inner and outer membrane regions when the ratio was 0.5. Furthermore, the membrane's porosity, water flux, and contact angle were all good.

**Author Contributions:** Conceptualization, M.H.D.O. and S.H.M.N.; methodology, S.H.M.N. and K.S.; validation, M.H.D.O., T.D.K. and K.Y.W.; formal analysis, T.D.K. and K.Y.W.; investigation, K.S.; resources, S.H.M.N.; data curation, K.S. and S.H.M.N.; writing—original draft preparation, K.S.; writing—review and editing, S.M., M.H.D.O. and R.K.; visualization, F.H.J.; supervision, M.H.D.O.; project administration, M.H.D.O.; funding acquisition, M.H.D.O. All authors have read and agreed to the published version of the manuscript.

**Funding:** The authors gratefully acknowledge financial support from the Ministry of Higher Education Malaysia for funding through the Higher Institution Centre of Excellence Scheme (project number: R.J090301.7809.4J430). The authors would also like to thank the JICA Technical Cooperation Project for ASEAN University Network/Southeast Asia Engineering Education Development Network (JICA Project for AUN/SEED-Net) via the Collaborative Education Program (project number: UTM CEP 2102a/R R.J130000.7309.4B651) and Universiti Teknologi Malaysia for research grants, namely the UTM Fundamental Research (UTMFR) (project number: Q.J130000.3809.22H07), Prototype Research

Grant-ICC (PRGS-ICC) (project number: R.J130000.7709.4J483), and Industry/International Incentive Grant (IIIG) (project number: Q.J130000.3609.03M17).

**Data Availability Statement:** The raw/processed data required to reproduce these findings cannot be shared at this time, as the data form part of an ongoing study.

**Acknowledgments:** The authors would also like to thank the Research Management Centre, Universiti Teknologi Malaysia, for technical support.

**Conflicts of Interest:** The authors declare no conflict of interest.

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
