# Peer review of "Photodegradation of Bisphenol a in Water via Round-the-Clock Visible Light Driven Dual Layer Hollow Fiber Membrane"

_catalysts, doi:10.3390/catal13050816_

Round 1
Reviewer 1 Report
The focus of this study has been the fabrication of a visible light-driven photocatalytic DLHF membrane. The incorporation of catalysts to hollow fiber membranes is very interesting for membrane reactors.
The work is original in the field of research in which it is framed. There are further doubts as to the possible practical application of the technology proposed for its application for the elimination of emerging pollutants in wastewater treatment plants.
A specific improvement that the authors should consider regarding the methodology is to conduct real-world experiments in wastewater treatment plants.
The conclusions are consistent with the evidence and arguments presented and do they address the main question posed.
The number of references could be higher.
Author Response
Dear Reviewer,
Thank you for taking the time to review our submission. We appreciate your feedback and suggestions for improving our work.
We have carefully considered your comments and have made several revisions to our manuscript. We have addressed the issues you raised and have provided additional information to support our claims.
Attached is the revised version of our manuscript. We hope that you find the changes we made satisfactory. Please let us know if you have any further comments or concerns.
Once again, we appreciate your time and effort in reviewing our submission. We look forward to hearing from you soon.
Best regards,

Reviewer 2 Report
This study is very close and similar to the previous work of the same group of authors (10.3390/membranes12020208). However, it is very strange that in their new manuscript, the authors do not mention that this study is a continuation of already published work. This significantly reduces the novelty and relevance of the presented work. Authors have significantly improve a revised manuscript before it will be accepted for publication. 1. The introduction section needs to be radically elaborated. Authors should demonstrate the state-of-the art of bisphenol A disposal problem and use up-to-date references. To the list of sources used - it is very limited and not modern and have to be expanded. 2. line 126 and henceforth use the abbreviation dual layer hollow fiber membrane. 3. subsection 2.4.1, 2.4.2, etc. - authors need to keep only information about the equipment used and remove well-known facts about used techniques. 4. In subsection 2.5.2, please add missing information about determining the concentration of BPA, its concentration in the experiment, testing temperature, etc. 5. Figure 3 - too small magnification. Figures 4-7 authors need to improve the quality of the figures 6. Authors have to explain why IR- and XRD spectra were presented only for one sample? it is necessary to provide similar data for the remaining samples. 7. Check the numbering of the figures. Figure #7 was repeated twice 8. Figure No. 7 (second one). The presented curves for the decomposition of BPA look very similar and the experimental data lie in the region of the standard deviation. The authors need to add standard deviation values to the graph. Based on the presented data, it is impossible to say unequivocally that sample (0.5) is a better catalyst than the other two samples. 9. The authors must calculate the value of the reaction rate constant for the decomposition of BPA. 10. Tables 2 and 5 should be improved or re-arranged into text format. In their present form they are not informative. Check table numbering (table No. 4 is missing) 11. Authors should provide comparative data on the effectiveness of the developed catalysts in comparison with previously published works, including their 2022 article. Our decision - major revision
Author Response
Dear Reviewer,
Please see attachment, thank you for taking the time to review our submission. We appreciate your feedback and suggestions for improving our work.
We have carefully considered your comments and have made several revisions to our manuscript. We have addressed the issues you raised and have provided additional information to support our claims.
Attached is the revised version of our manuscript. We hope that you find the changes we made satisfactory. Please let us know if you have any further comments or concerns.
Once again, we appreciate your time and effort in reviewing our submission. We look forward to hearing from you soon.
Best regards,
Khalis

Round 2
Reviewer 2 Report
First of all, the authors still need to respond to my comment about this paper's strong similarity with their previous paper (10.3390/membranes12020208). Moreover, no explanations were added to the introduction section. Thus the scientific novelty still needs to be higher. This point is very critical and needs to be responded to by authors.
1. Line #82 - please correct the reference (Saravanan et al., 2017) to the MDPI format. The same query for line #461
2. I still need more response from the Authors for my query #6 about the presence of XRD patterns of other studied samples. The authors claimed they focused on the 0.5 Cu2O/WO3/PVDF sample. However, a higher loading degree of the Cu2O/WO3 part can dramatically affect the crystallinity degree of composites... I am still asking again to add appropriate XRD patterns to the supplementary file. Also, please recognize all peaks on the XRD pattern presented in the manuscript.
Based on the above, I suggest a major revision for a more thorough revision of the introduction section et al.
